# Intracranial Metastases from Uterine Leiomyosarcoma: A Systematic Review and Case Illustration

**DOI:** 10.3390/jcm14186631

**Published:** 2025-09-20

**Authors:** Ahmad Pour-Rashidi, Sara Zandpazandi, Laetitia Perronne, Virginia B. Hill, Chase Krumpelman, Kamal Subedi, Linda Kelahan, Amir A. Borhani, Hatice Savas, Ryan Avery, Tugce Agirlar Trabzonlu, Ulas Bagci, Sean Sachdev, Karan Dixit, Rimas V. Lukas, Priya Kumthekar, Yuri S. Velichko

**Affiliations:** 1Department of Radiology, Feinberg School of Medicine, Northwestern University, Chicago, IL 60611, USA; ahmad.pourrashidi@northwestern.edu (A.P.-R.); laetitia.perronne@northwestern.edu (L.P.); virginia.hill@nm.org (V.B.H.); chase.krumpelman@nm.org (C.K.); kamal.subedi@nm.org (K.S.); linda.kelahan@nm.org (L.K.); amir.borhani@nm.org (A.A.B.); hatice.savas@nm.org (H.S.); ryan.avery@nm.org (R.A.); tugce.agirlartrabzonlu@nm.org (T.A.T.); ulas.bagci@northwestern.edu (U.B.); 2Department of Neurological Surgery, New York-Presbyterian Hospital Weill Cornell Medicine, New York, NY 10065, USA; saz4008@med.cornell.edu; 3Robert H. Lurie Comprehensive Cancer Center, Northwestern University, Chicago, IL 60611, USA; sean.sachdev@northwestern.edu (S.S.); karan.dixit@northwestern.edu (K.D.); rimas.lukas@northwestern.edu (R.V.L.); priya.kumthekar@nm.org (P.K.); 4Department of Radiation Oncology, Feinberg School of Medicine, Northwestern University, Chicago, IL 60611, USA; 5Department of Neurology, Feinberg School of Medicine, Northwestern University, Chicago, IL 60611, USA; 6Department of Neurological Surgery, Feinberg School of Medicine, Northwestern University, Chicago, IL 60611, USA; 7Northwestern Medicine Malnati Brain Tumor Institute of the Lurie Comprehensive Cancer Center, Feinberg School of Medicine, Northwestern University, Chicago, IL 60611, USA

**Keywords:** Leiomyosarcoma, brain metastasis, surgery, radiotherapy, survival, systematic review

## Abstract

**Background/Objectives**: Brain metastasis from uterine leiomyosarcoma (ULMS) is an exceptionally rare complication of an aggressive malignancy. With fewer than 40 cases previously documented, a significant knowledge gap exists regarding its clinical course, management, and outcomes. This study provides the largest analysis of ULMS brain metastases to date, integrating a systematic literature review with a novel case report illustrating the disease’s uniquely rapid progression. **Methods**: Following PRISMA guidelines, we systematically reviewed four major databases to identify all reported cases of intracranial metastasis from ULMS. Data on patient demographics, clinico-radiological features, treatments, and survival were extracted and analyzed. Methodological quality was assessed using a modified Joanna Briggs Institute (JBI) tool. **Results**: We analyzed 34 studies with 39 individual cases. Additionally, this review was supplemented by one new illustrative case from our institution. The median patient age was 51.5 years, and most presented with focal neurological symptoms. Common imaging findings included hyperdense lesions on CT and homogeneously enhancing, dural-based masses on MRI, which mimic other intracranial pathologies. Though surgery was the most frequent intervention (76.9%), median survival after a brain metastasis diagnosis was a grim 5 months, with no significant difference observed between treatment modalities. Our illustrative case was remarkable for an extremely rapid volumetric doubling time averaging just 7.3 days. **Conclusions**: Brain metastasis from ULMS is a lethal event with an extremely poor prognosis. Nonspecific imaging features create diagnostic challenges, necessitating histopathological confirmation. Current therapies, including surgery and radiotherapy, offer palliative benefit but do not significantly alter survival. The aggressive biological behavior demonstrated here underscores the urgent need for increased clinical awareness and collaborative research to develop more effective management strategies and improve outcomes for this devastating diagnosis.

## 1. Introduction

Leiomyosarcoma (LMS) is an aggressive malignant tumor that originates from smooth muscle cells and belongs to the heterogeneous group of soft tissue sarcomas [1]. LMS includes several distinct subtypes, primarily differentiated by their anatomical site of origin, which dictates their biological behavior and clinical outcomes [2]. The primary subtypes include uterine LMS, retroperitoneal LMS, and LMS of the soft tissues of the extremities [1]. These variants display significant heterogeneity in their rates of growth, metastatic potential, and response to therapy. Key prognostic factors across all subtypes include tumor size, histological grade, anatomic location, tumor stage, and the feasibility of complete surgical resection [3].

A defining characteristic of LMS is its propensity for hematogenous metastasis [4]. The lungs are the most common site of distant spread, followed by the liver, particularly for LMS arising in the abdomen or uterus. However, the risk of metastasis varies substantially among subtypes [5]. For instance, cutaneous LMS has a much lower risk of spreading (5% to 10%) whereas more aggressive forms like uterine, vascular, and deep soft tissue LMS (especially retroperitoneal ones) have a much higher risk, around 30% to 40% or even more [3].

Uterine leiomyosarcoma (ULMS) is the most prevalent type of uterine sarcoma, although it accounts for only 1–2% of all uterine cancers. Like other LMS subtypes, ULMS typically metastasizes via the bloodstream, with the lungs, peritoneal and retroperitoneal cavities, and bones being the most frequent secondary sites [6]. The occurrence of brain metastases from ULMS is exceptionally rare, with only approximately 40 cases documented in the literature to date. The rarity of this complication has resulted in a scarcity of literature to guide clinical management [7].

This knowledge gap underscores the importance of a comprehensive analysis of reported instances to better understand the disease course in these patients. Such an evaluation is essential for clarifying the timeline to the development of brain metastases, the patterns of intracranial spread (e.g., solitary versus multiple lesions), and associated clinical presentations. Furthermore, a detailed review of these uncommon cases is essential for evaluating the effectiveness of various treatments, including surgery, radiation therapy (whole-brain or stereotactic), and systemic agents.

In this study, we present a comprehensive analysis of documented cases of ULMS with brain metastases, with a focus on imaging characteristics, treatment strategies, and survival outcomes. We also present a novel case that illustrates previously unreported tumor features, further highlighting the aggressive nature of ULMS brain metastases.

## 2. Materials and Methods

### 2.1. Study Design and Patient Selection

This retrospective study was approved by the Institutional Review Board and received waivers for both Health Insurance Portability and Accountability Act (HIPAA) authorization and written informed consent. The institutional electronic data warehouse (EDW) was searched from January 2004 to December 2022 to identify patients meeting the following inclusion criteria: diagnosis of biopsy-proven Soft Tissue Sarcoma and pre-treatment MRI of the primary tumor. During the dataset review, a single ULMS patient with brain metastases was identified. Patient confidentiality was strictly maintained, and all identifying information was anonymized.

### 2.2. Literature Search and Review Protocol

This systematic review was conducted and reported in accordance with the Preferred Reporting Items for Systematic Reviews and Meta-Analysis (PRISMA-2020) guidelines. The review was designed to address a research question developed using the PICO (Population, Intervention, Comparison, Outcome) framework [8]. Specifically, the review aimed to answer the following PICO-based question: In patients with ULMS, what are the clinical features, imaging findings, and outcomes associated with intracranial metastases, and how do different treatment modalities compare in terms of survival?

Systematic search was performed in PubMed, Google Scholar, Embase, and Web of Science using a predefined search strategy, with the final search conducted on 1 May 2025. The search strategy combined relevant keywords with medical subject headings (MeSH), where applicable. Search terms included combinations of: ((((((((((leiomyosarcoma) AND (brain)) OR (cerebral)) OR (intracranial)) OR (intracerebral))) OR (CNS)) OR (central nervous system)) AND (metastasis))) OR (metastatic). All the terms were searched in the title, abstracts, or keywords of all records (Appendix A).

### 2.3. Inclusion Criteria

Only retrospective case reports, case series, and other studies were identified for this review. Moreover, all eligible studies were enrolled in this study with no time and language restrictions.

### 2.4. Exclusion Criteria

Relevant exclusion criteria were: (1) diagnosis of any other sarcoma, (2) data were only published in abstract form, and (3) studies with incomplete data.

### 2.5. Data Search and Extraction

Two independent reviewers read the drafted articles in full text and extracted the following data: first author’s name, year of study, location of study, age, presenting symptoms, location of brain metastasis, tumor consistency, imaging features, contemporary lung metastasis, treatments, extent of resection in case of surgery, time from primary tumor to metastasis, survival after diagnosis of metastasis, cause of death, and immunohistochemistry characteristics. In case of any disagreement between the two reviewers, a third one made the final decision.

### 2.6. Quality Assessments and Analysis

Data extraction was independently carried out by two reviewers and encompassed a detailed assessment of study characteristics, patient demographics, common clinical features, radiological and histopathological findings, as well as follow-up data. To assess methodological quality and potential risk of bias, the included studies were evaluated using a modified version of the Joanna Briggs Institute (JBI) critical appraisal checklist for case reports, as previously described in the literature [9]. Discrepancies in data extraction or quality assessment were resolved through consensus or consultation with a third reviewer when necessary.

All statistical analyses were conducted using IBM SPSS Statistics version 29.0.2.0. Given the heterogeneity among the included studies, quantitative analysis was limited to descriptive statistics, including the calculation of means and ranges for relevant variables. As a result, data synthesis was predominantly qualitative in nature.

## 3. Results

### 3.1. Search Results and Study Characteristic

The systematic literature search, conducted using the predefined strategy, initially identified 2170 records. After removing 538 duplicates, 1632 unique articles remained for screening. Title and abstract screening excluded 1593 articles based on irrelevance to the research question. The full texts of the remaining 39 studies were assessed for eligibility. The most common reasons for exclusion at this stage were insufficient data (4 studies) and inability to access the full text (1 study). Following the application of all inclusion and exclusion criteria, 34 studies with a total of 39 individual cases were deemed eligible and included in the final analysis (Figure 1).

### 3.2. Patient Characteristics

From 1989 to 2025, 39 individual cases of brain metastases from uterine leiomyosarcoma (ULMS) were identified in 34 previously published studies. We also included one new case from our institution that met all inclusion criteria, bringing the total number of cases analyzed to 40. As expected, all patients were female, with a median age of 51.5 years (range: 26–70) (Table 1). For patients who were still alive at the time of data collection, survival was calculated to the last known follow-up date. This right-censoring of data is a key limitation, as it inherently underestimates the true survival time for this subgroup and contributes to a poorer overall prognostic outcome. Four patients were excluded from the survival analysis because their survival time was not reported or could not be estimated. Therefore, for the remaining 36 patients, the average and median survival times after diagnosis of brain metastasis were 10.37 months and 5 months, respectively.

### 3.3. Presentation Symptoms

All patients exhibited neurological symptoms corresponding to the anatomical location of the metastatic lesion, typically at the gray-white matter junction. Of the cases reviewed, 30 patients (75.0%) had a single brain lesion, 5 patients (12.5%) presented with two lesions, and another 5 patients (12.5%) had more than two intracranial lesions. Concurrent pulmonary metastases were documented in 29 out of 35 cases (82.86%) where lung involvement was reported. The median interval from initial diagnosis of the primary uterine tumor to the development of brain metastases was 31 (range 0–116) months (Table 1).

### 3.4. Radiological Features and Diagnosis

Nearly all patients underwent brain MRI (T1- and T2-weighted sequences), with or without accompanying CT imaging. Non-contrast CT scans, when performed, commonly revealed hyperattenuating lesions or evidence of osteolytic changes. Radiological findings were reported in 23 studies. Among MRI signal characteristics, T1 and T2 isointensity was observed in 4 (17.4%) cases, T2 hyperintensity in 5 (21.7%) cases, and a combination of T1 isointensity with T2 hyperintensity in 3 (13%) cases. Homogeneous contrast enhancement was the most frequently reported pattern, seen in 19 (82.6%) cases, while ring enhancement was described in 4 (17.4%) cases. Additional MRI features included necrosis in 14 (60.9%) cases and a dural-based lesion in 10 (43.5%) cases. (Appendix A).

### 3.5. Histopathology

Sixteen studies described histopathological features, consistently noting the presence of firm, moderately vascularized tissue with intermittent hemorrhagic areas. Immunohistochemical analysis was reported in 15 cases, with positive staining observed for actin in 12 (80%) cases, vimentin in 8 (53.3%) cases, and desmin in 8 (53.3%) cases (Table 1). While a comprehensive marker panel was often unavailable, the Ki-67 proliferation index was specifically reported in two instances [24,30], with values of approximately 20% and greater than 25%.

### 3.6. Treatment Approaches and Outcomes

Of the reported cases, 30 (76.9%) patients underwent surgical intervention, either alone or in combination with adjuvant therapies. GKRS, with or without chemotherapy, was administered to 5 (12.8%) patients, while 3 (7.7%) patients received whole-brain radiotherapy with or without chemotherapy. One (2.6%) patient was treated solely with chemotherapy (Table 1). Follow-up information was available for 35 subjects. Our new case was treated with Gamma Knife radiosurgery and chemotherapy. The median survival following the diagnosis of brain metastases was 5 months. At the time of publication, 10 patients (28.6%) were reported to be alive (Table 1).

To evaluate the impact of different treatment strategies on survival, patients were stratified into six groups (Table 2). A Kaplan–Meier plot visually demonstrates the survival probabilities for each treatment group over time (Figure 2). While an overall log-rank comparison of the groups did not show a statistically significant difference (*p* = 0.29), a specific pairwise test revealed a notable treatment effect: patients treated with surgery and chemotherapy had a significantly longer median survival (18.0 months) compared to those who received surgery alone (4.0 months, *p* = 0.03). No other pairwise comparisons between these treatment groups reached statistical significance.

Further analysis isolating the effect of chemotherapy confirmed its benefit. Patients who received chemotherapy as part of any treatment regimen had a median survival more than two times longer than those who did not (12 vs. 5 months). This finding was statistically significant (*p* = 0.04), strongly suggesting that including chemotherapy in the treatment plan is associated with improved survival for this patient cohort. A Cox proportional-hazards model further quantified the benefit of chemotherapy. It demonstrated that adding chemotherapy to a regimen reduced the hazard of death by approximately 70% compared to those without it. This finding reinforces that systemic therapy is a critical component of treatment for this aggressive disease.

### 3.7. Methodological Quality and Risk of Bias Assessments

A modified form of the JBI case series appraisal tool was employed to assess the methodological quality of the included studies (Table 3). The mean (SD) score was 7.1 (SD 1.03), reflecting good methodological quality. Lower-scoring studies lacked detailed pathology reports or follow-up assessments for patients and gave limited details on patient demographic characteristics and individual outcomes. The highest-scoring study offered a more complete account of the disease trajectory and patient follow-up. Nevertheless, the small sample sizes across the case series limit the broader applicability of the findings (Table 3. Additional details in Appendix A).

### 3.8. Illustrative Case

A 49-year-old female with a history of myomectomy 14 years prior, presented with right arm paresis, two months after being diagnosed with ULMS with concurrent lung metastases. Initial brain computed tomography (CT) revealed multiple hyperdense lesions with surrounding edema (Figure 3). A subsequent diagnostic magnetic resonance imaging (MRI) identified a total of four intracranial lesions, which appeared T1-weighted hypointense and T2-weighted hyperintense with homogeneous enhancement and central necrosis (Figure 4). The three largest lesions measured 11.8 × 11.2 × 11.3 mm, 22.6 × 17.6 × 21.3 mm, and 21.1 × 18.8 × 19.4 mm, with a mean Apparent Diffusion Coefficient (ADC) of 973 ± 37 × 10^−6^ mm^2^/s.

Given the findings, the patient was scheduled for GKRS. However, a pre-treatment planning MRI obtained just seven days later revealed aggressive disease progression. The total lesion count had increased to six, and the three largest lesions demonstrated a mean volumetric growth of 94% (range: 51–141%), corresponding to a mean volumetric doubling time of only 7.3 days (range: 5.5–11.7 days) (Figure 5) [42].

The patient proceeded with GKRS using frame-based immobilization. All lesions were treated to a median dose of 20 Gy, prescribed to the 50% isodose line, with 100% coverage of all visible and defined tumors. A two-month follow-up MRI confirmed a successful response in the targeted lesions, which showed a mean volumetric decrease of 73% (range: 62–86%) and a corresponding increase in the mean ADC to 2344 ± 270 × 10^−6^ mm^2^/s. Despite this effective local therapy, the development of new metastases was evident, with the total intracranial lesion count rising to twelve, signifying ongoing overall disease progression (Figure 5). Unfortunately, the patient ultimately succumbed to her systemic disease burden 20 months after the diagnosis of brain metastases.

## 4. Discussion

Brain metastasis from ULMS is an exceptionally rare and ominous complication of a disease already known for aggressive behavior and hematogenous spread. This analysis represents the largest collection of such cases to date, providing valuable insights into the clinical presentation, radiological features, treatment patterns, and outcomes associated with this disease entity. These key findings emphasize the aggressiveness of ULMS when it involves the brain, a median survival of just five months after diagnosis, underlining the serious diagnostic and therapeutic challenges.

A critical finding from our review is the radiological profile of these metastases. The frequent presentation as homogeneously enhancing, dural-based masses on MRI and hyperdense lesions on non-contrast CT creates a significant diagnostic challenge [4,6,43]. These features can, respectively, mimic more common intracranial pathologies such as meningiomas, particularly in patients without a known primary cancer, and hemorrhagic metastases from other primary sites like melanoma, renal cell carcinoma, or choriocarcinoma [4,6,43]. This radiological ambiguity underscores the need for a high index of suspicion. Histopathological confirmation is critical for an accurate diagnosis. The high prevalence of concurrent pulmonary metastases (82.86%) in our cohort suggests that in a ULMS patient with known lung involvement, the development of new neurological symptoms should immediately prompt investigation for intracranial spread.

The management of ULMS brain metastases remains a significant challenge with no established standard of care, a fact reflected in the variety of treatments employed in the reported cases [13,41]. Surgery was the most common intervention (76.9%), often utilized for diagnostic confirmation, cytoreduction, and palliation of mass effect. While a broad comparison of treatment modalities did not show an overall survival benefit, a more specific pairwise analysis revealed a critical difference: patients treated with surgery combined with chemotherapy had a significantly longer median survival than those treated with surgery alone (18.0 vs. 4.0 months, *p* = 0.03). This was the only treatment comparison to reach statistical significance. Furthermore, a focused analysis confirmed that the inclusion of chemotherapy in any regimen was the key factor associated with improved survival (*p* = 0.04). In contrast, a focused analysis revealed that the inclusion of chemotherapy was the only factor associated with significantly longer survival (18.0 vs. 5.0 months, *p* = 0.04), suggesting its critical role in improving outcomes [44,45]. A Cox proportional-hazards model reinforced this conclusion, showing that the addition of chemotherapy was associated with a roughly 70% reduction in the hazard of death compared to regimens without it. This finding underscores that achieving systemic control is likely the most critical component of the treatment strategy for this aggressive disease. This lack of a clear survival advantage may be attributable to several factors, including the small sample size, inherent selection bias in retrospective case reports (e.g., patients with better performance status are more likely to be offered surgery), and the overwhelming impact of systemic disease burden. The grim median survival of 5 months is unambiguously poor, even when compared to brain metastases from other aggressive cancers, suggesting an inherent biological aggressiveness once the central nervous system is involved.

Our case strikingly illustrates the clinical course and challenges associated with these tumors. The patient’s rapid radiological progression in just one week exemplifies their aggressive biological nature. Notably, her 20-month survival following Gamma Knife radiosurgery significantly exceeds the 4-month median observed in our review, suggesting that aggressive local therapy can achieve durable local control in select patients. As mentioned, the lesion doubled in size and, more critically, the number of lesions markedly proliferated within a single week, underscoring a concerning lack of response to the targeted therapy. This highlights the potential importance of initiating treatment as early as possible. Given this scant data, performing precise, regular neurologic examinations and obtaining brain imaging even with minimal neurologic changes may be of utmost importance to improve patient care. Conversely, our review of published cases indicates no significant difference in outcomes among various approaches to these metastases. This suggests that while local control of intracranial disease is crucial for neurological palliation, it may not alter the overall fatal course driven by widespread metastases. Ultimately, the prognosis appears to be dictated by the systemic tumor burden and its response to therapy.

This study is not without limitations. The foremost is its reliance on published case reports and small case series, which are inherently susceptible to publication bias (e.g., favoring the reporting of unusual or successful outcomes) and heterogeneity in reporting standards, treatment protocols, and follow-up duration across several decades. The small sample size (n = 40) limits the statistical power to detect significant differences between treatment groups and draw definitive conclusions about the optimal therapeutic strategy. The methodological quality assessment, while generally good, did reveal variability in the completeness of reporting, further constraining the depth of analysis.

Despite these limitations, our review offers important clinical insights. We recommend that clinicians maintain a high index of suspicion for brain metastases in ULMS patients, particularly those with confirmed pulmonary metastases, as this is a common pathway for disease spread. We propose a strategy of close, regular neurological examinations for this high-risk subset of patients. Furthermore, we advise that brain imaging, such as an MRI, be performed promptly at the first sign of even minimal neurological changes. This proactive approach, while not a universal screening program, could significantly reduce diagnostic delays and may allow for earlier intervention, potentially improving patient outcomes. A multidisciplinary approach is crucial to tailor therapy to the individual patient. This involves neurosurgeons, radiation oncologists, and medical oncologists who consider factors like the number and location of metastases, systemic disease status, and overall performance. Future research should focus on establishing a prospective, multicenter registry for rare metastases like these to enable more robust analysis. Furthermore, molecular and genomic profiling of these metastatic tumors is warranted to identify potential therapeutic targets that could lead to more effective systemic agents.

## 5. Conclusions

In conclusion, brain metastasis from ULMS is a rare but lethal event characterized by nonspecific imaging features and an extremely poor prognosis. While current treatments, including surgery and radiotherapy, may offer symptomatic benefits, they do not appear to significantly alter the grim overall survival. Increased awareness and collaborative, prospective research are urgently needed to develop more effective management strategies and improve outcomes for this patient population.

## Figures and Tables

**Figure 1 jcm-14-06631-f001:**
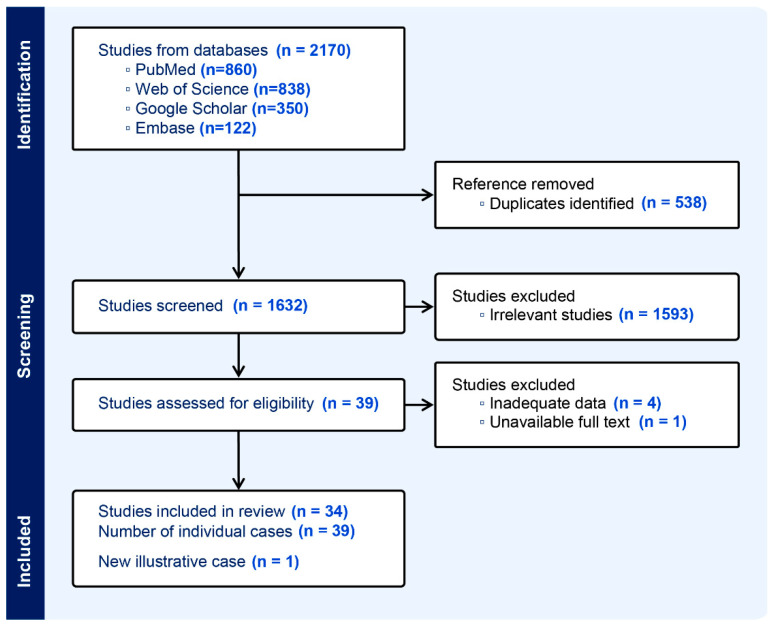
PRISMA diagram.

**Figure 2 jcm-14-06631-f002:**
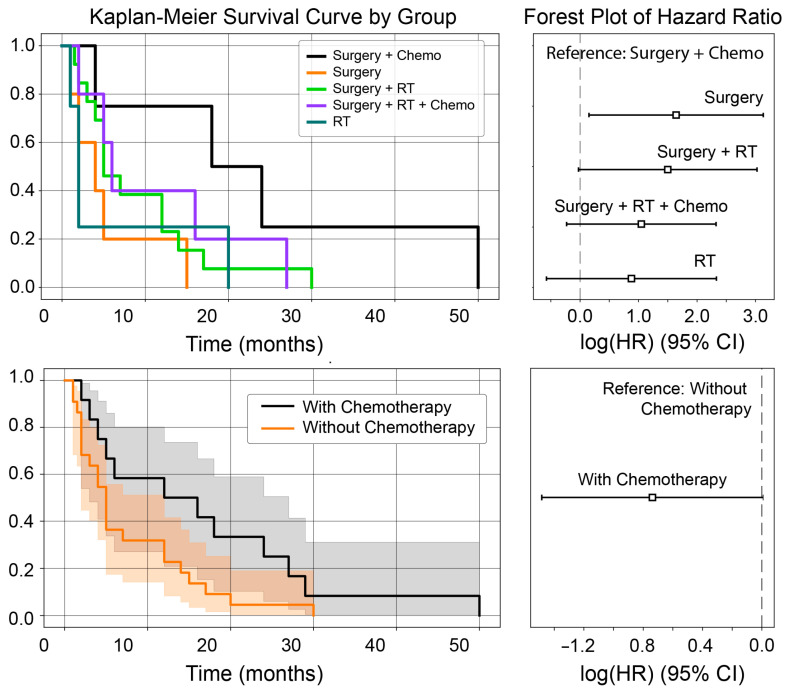
Kaplan–Meier survival curves illustrating the probability of survival over time for patients in different treatment groups. A Cox proportional-hazards model was used to demonstrate the relative hazard of death for each group.

**Figure 3 jcm-14-06631-f003:**
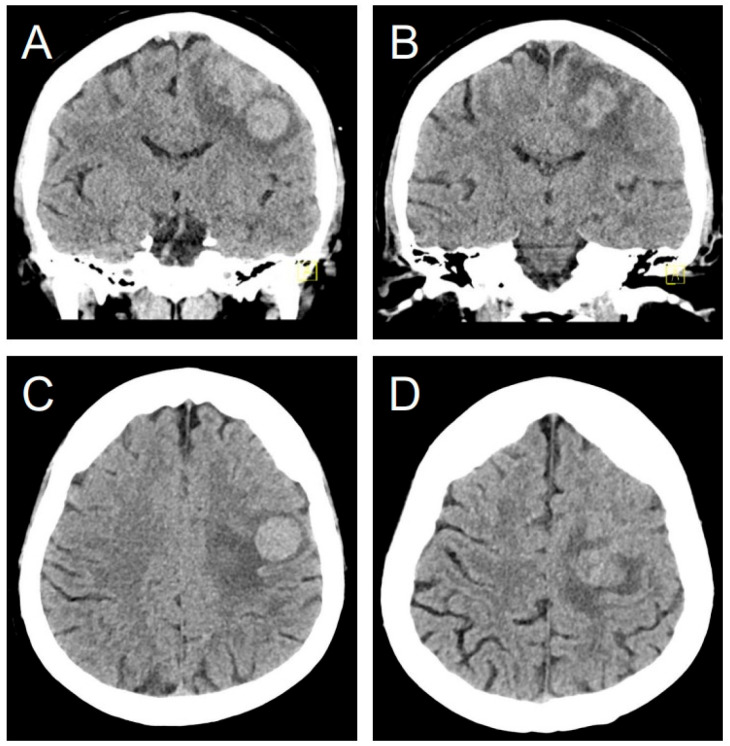
Pretreatment non-contrast brain CT scan in coronal (**A**,**B**) and axial (**C**,**D**) views shows a hyperdense left posterior frontal lobe lesion with a marked vasogenic edema.

**Figure 4 jcm-14-06631-f004:**
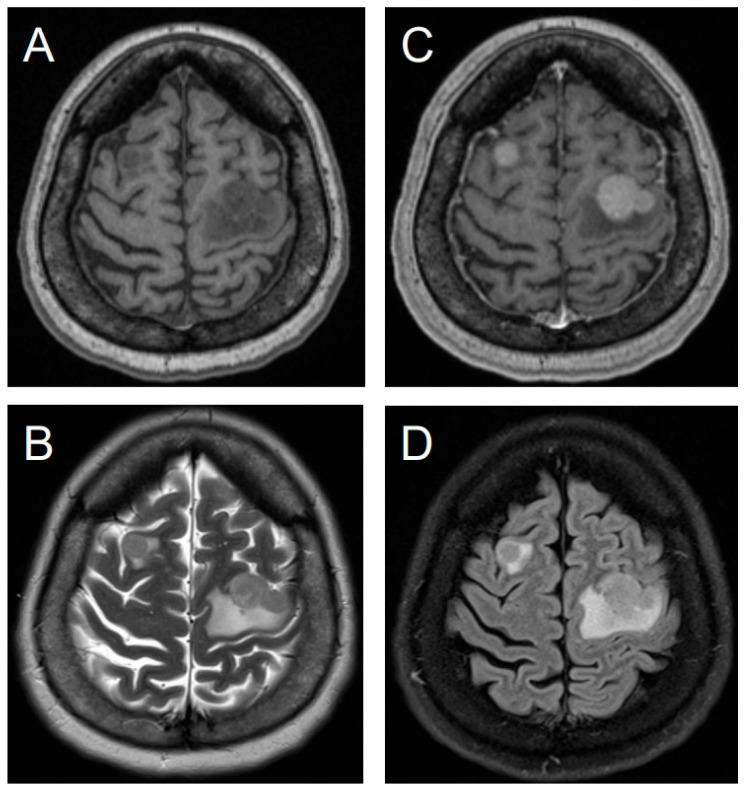
Pretreatment axial non-contrast T1-weighted (**A**) and T2-weighted (**B**) brain MRI demonstrate two lesions, hypointense on T1W and hyperintense on T2W, located in the right and left frontal lobes, which show homogeneous enhancement on post-contrast T1W imaging (**C**) and are surrounded by extensive vasogenic edema on FLAIR (**D**).

**Figure 5 jcm-14-06631-f005:**
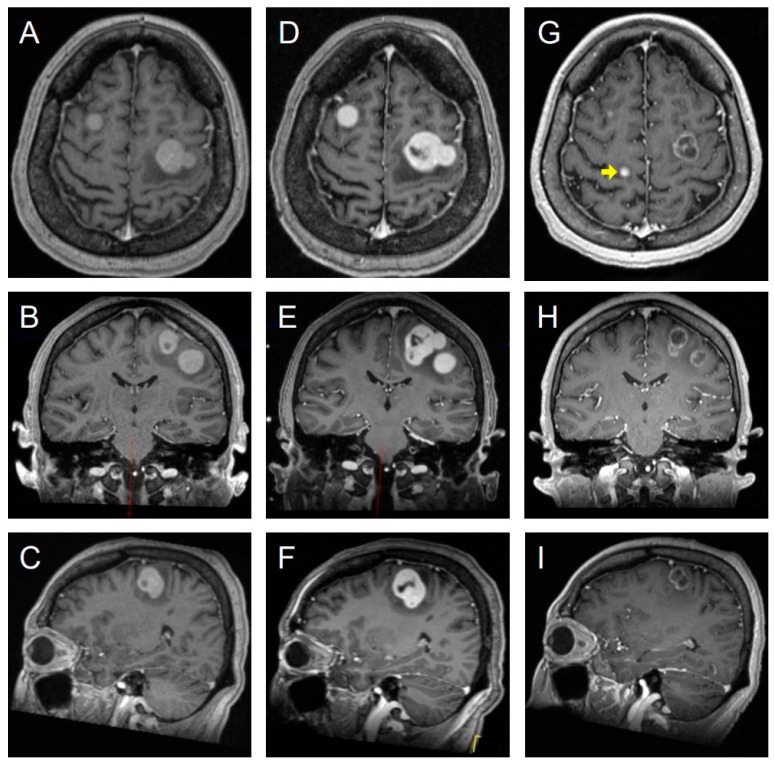
Post-contrast T1-weighted brain MRIs: Axial, coronal, and sagittal views across three time points. Columns (**A**–**C**) represent the initial scan at admission. Columns (**D**–**F**), taken one week later, demonstrate rapid lesion progression. Columns (**G**–**I**) show the tumor’s response two months post- GKRS, highlighting an acceptable reduction in existing lesions despite the concurrent emergence of a new lesion (yellow arrow).

**Table 1 jcm-14-06631-t001:** Patients’ characteristics.

Author	Age	Lung Mets	Time to BM	EOR	Adjuvant for BM	Positive IHC Markers	Cause of Death	Survival After BM, Month
Vaquero et al. [10]	26	Yes	NS	T	Chemo	NS	CNS	50
Prussia et al. [11]	36	NS	72	T	No	NS	NS	NS
Wronski et al. [12]	60	Yes	78	T	WBRT	NS	Systemic	30
Bindal et al. [13]	55	Yes	116	T	WBRT	NS	Systemic	14
Uchino et al. [14]	54	No	48	T	Chemo	NS	Lung	24
Salvati et al. [4]	58	Yes	3	T	WBRT	NS	NS	3
	43	Yes	48	T	WBRT	NS	NS	5
	28	Yes	26	T	WBRT	NS	NS	5
Ziyal et al. [15]	38	Yes	72	T	WBRT	NS	CNS	4
Mawrin et al. [16]	59	Yes	36	Bx	WBRT	Laminin, Vimentin	CNS	2
Yip et al. [17]	63	NS	15	T	No	Actin, Desmin	Systemic	4
Munakata et al. [18]	52	No	36	Bx	GKRS	NS	NS	NS
Melone at al. [19]	57	Yes	12	T	WBRT + Chemo	NS	___	At least 5
Kaya et al. [20]	60	Yes	0	___	WBRT + Chemo	Actin, Vimentin, Desmin	NS	NS
Benizelos et al. [21]	51	Yes	6	___	WBRT + Chemo	Actin, Desmin	Systemic	3
Pereira et al. [22]	55	No	60	T	No	Actin, Desmin	Systemic	5
Honeybul et al. [23]	42	Yes	10	T	No	NS	Systemic	2
Venizelos et al. [24]	57	Yes	8	STR	WBRT	Actin, Vimentin, Desmin	Systemic	1.5
Yamada et al. [25]	50	Yes	28	STR	GKRS	NS	Systemic	12
Mariniello et al. [26]	57	No	8	STR	Chemo	Actin, Vimentin	CNS	4
Chen et al. [27]	54	Yes	48	___	GKRS + Chemo	NS	___	At least 29
Shepard et al. [28]	54	NS	24	___	GKRS	NS	Systemic	1
Gurram et al. [29]	59	Yes	84	STR	SRS	NS	Systemic	2
Abrahao et al. [30]	45	Yes	31	T	WBRT + Chemo	NS	NS	27
Abrahao et al. [30]	51	Yes	22	T	WBRT + Chemo	NS	Systemic	16
Kim et al. [31]	57	Yes	36	___	____	___	Systemic	1
Inoue et al. [32]	48	Yes	77	T	Chemo	Actin, EMA, Vimentin, Desmin	___	At least 18
Ahuja et al. [33]	60	Yes	6	___	Chemo	Actin, Vimentin, pan-CK	___	At least 12
Sosa et al. [34]	43	Yes	0	T	No	Actin	Systemic	1
Chahdi et al. [35]	46	NS	84	T	WBRT	Actin, h-Caldesmon	NS	NS
Imoumby et al. [36]	46	Yes	60	T	WBRT	Actin, Vimentin, h-Caldesmon	Systemic	5
Miki et al. [37]	35	NS	36	T	WBRT	NS	___	At least 17
	70	Yes	96	T	No	NS	___	At least 15
	51	Yes	48	T	WBRT	NS	___	At least 7
Soo et al. [38]	60	No	0	T	WBRT + Chemo	NS	___	At least 6
Delgado et al. [39]	33	Yes	10	T	WBRT + Chemo	NS	NS	NS
Eatz et al. [40]	55	Yes	5	Bx	SRS	NS	Systemic	2
Richards et al. [41]	51	Yes	44	T	Chemo + SRS	Myosin	___	At least 2
Carrilo-Uzeta et al. [7]	46	No	0	T	WBRT	SMA, h-Caldesmon	___	At least 12
Our Case	49	Yes	2	___	GKRS + Chemo	Actin, Vimentin	Systemic	20

Mets: metastasis; EOR: extent of resection; NS: not specified; SMA: smooth muscle antigen; EMA: epithelial membrane antigen; Chemo: chemotherapy; WBRT: whole brain radiotherapy; GKRS: Gamma Knife radiosurgery; Bx: biopsy; T: total tumor removal; STR: subtotal tumor removal.

**Table 2 jcm-14-06631-t002:** Survival times for different therapeutic options.

Treatment	Number of Patients	Mean Survival Time	Median Survival Time
Surgery + Chemotherapy	4	24 ± 19.2	18 (24–18)
Surgery	5	5.4 ± 5.6	4 (5–2)
Surgery + RT	13	9.3 ± 8	5 (12–4)
Surgery + RT + Chemotherapy	5	11.2 ± 10.3	6 (16–5)
RT	4	6.2 ± 9.2	2 (2–2)
With Chemotherapy	12	15.1 ± 13.9	12 (24–5)
Without Chemotherapy	23	7.7 ± 7.4	5 (12–2)

**Table 3 jcm-14-06631-t003:** The JBI Case Reports Appraisal Scores.

Case	Total Score (out of 8)	Case (Continuation)	Total Score (out of 8)
Vaquero et al., 1989 [10]	6	Honeybul et al., 2009 [23]	8
Prussia et al., 1992 [11]	4.5	Venizelos et al., 2011 [24]	8
Wronski et al., 1994 [12]	7	Yamada et al., 2011 [25]	8
Bindal et al., 1994 [13]	7	Mariniello et al., 2012 [26]	8
Uchino et al., 1996 [14]	7.5	Sosa et al., 2018 [34]	8
Salvati et al., 1998 [4]	7	Chahdi et al., 2018 [35]	5
Ziyal et al., 1999 [15]	7.5	Kim et al., 2016 [31]	5.5
Mawrin et al., 2002 [16]	8	Eatz et al., 2023 [40]	7
Yip et al., 2006 [17]	7.5	Soo et al., 2022 [38]	7.5
Chen et al., 2013 [27]	7	Carrillo-Uzeta et al., 2025 [7]	8
Shepard et al., 2014 [28]	6	Abrahao et al., 2015 [30]	7
Gurram et al., 2014 [29]	8	Delgado et al., 2022 [39]	6
Munakata et al., 2006 [18]	5	Inoue et al., 2016 [32]	8
Melone et al., 2008 [19]	8	Ahuja et al., 2017 [33]	8
Kaya et al., 2009 [20]	6	Miki et al., 2021 [37]	7.5
Benizelos et al., 2009 [21]	7	Richards et al., 2023 [41]	8
Pereira et al., 2011 [22]	8	Imoumby et al., 2021 [36]	8

## Data Availability

Data are contained within the article and Appendix A.

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
