# Peer review of "Intracranial Metastases from Uterine Leiomyosarcoma: A Systematic Review and Case Illustration"

_jcm, 2025, doi:10.3390/jcm14186631_

Round 1

Reviewer 1 Report

Comments and Suggestions for Authors

Manuscript entitled “Intracranial Metastases from Uterine Leiomyosarcoma: A Systematic Review and Case Illustration” by Ahmad Pour-Rashidi et al.

This study tackles a clinically important—but extremely uncommon—problem: brain metastasis from uterine leiomyosarcoma (ULMS). By combining a PRISMA-guided systematic review with an illustrative case displaying explosive intracranial growth, the authors deliver the largest synthesis to date.

Comments:

  1. To ensure transparency and reproducibility of the literature search, the authors should provide a search-strategy appendix detailing the exact search strings used for each database, including the date range, controlled vocabulary terms (e.g., Emtree or MeSH), Boolean operators, and any applied language or publication type filters.
  2. There is a discrepancy between the text, which states that 39 articles were reduced to 34 reports, and the table, which still lists 39 literature cases. The authors should clarify whether this reduction reflects the merging of multiple publications referring to the same patient and update the table accordingly to ensure consistency and accurate representation of the dataset.
  3. The manuscript would benefit from either a Kaplan–Meier pooled survival analysis, if individual patient data are accessible, or the inclusion of a forest plot comparing survival outcomes across different treatment modalities such as surgery, radiosurgery, and WBRT with or without chemotherapy.
  4. The inclusion of immunohistochemical panel positivity percentages adds value, yet the absence of key proliferative or molecular markers such as Ki-67, PIK3CA, or TP53 limits the depth of tumor characterization. If Ki-67 data were reported in any of the cases, the authors should include its range.
  5. Tables 1 and 2 are overly large, spanning multiple pages with over 40 columns in a font size that compromises readability, and the dense use of abbreviations further impedes clarity. The authors should restructure the data presentation by providing (i) a concise summary table outlining aggregate characteristics such as demographics and counts of key clinical or pathological variables, and (ii) a detailed per-patient table as a supplementary file. All figures should be exported at a minimum of 300 dpi, use a color-blind-safe palette, and employ vector graphics to ensure publication-quality resolution and accessibility.
  6. The manuscript contains numerous typographical errors (e.g., “symptems,” “inlcudes”), inconsistent verb tenses, and excessively long sentences, many exceeding 40 words, which collectively hinder readability. A professional copy-edit is necessary to correct these issues and improve overall fluency. The authors should aim to limit sentence length to 25 words or fewer to enhance clarity.
  7. While the flow diagram in Figure 1 adequately outlines the study selection process, the accompanying PRISMA 2020 checklist is missing. To comply with current reporting standards and ensure methodological transparency, the authors should complete and upload the PRISMA 2020 checklist as a supplementary file.

Author Response

C1: To ensure transparency and reproducibility of the literature search, the authors should provide a search-strategy appendix detailing the exact search strings used for each database, including the date range, controlled vocabulary terms (e.g., Emtree or MeSH), Boolean operators, and any applied language or publication type filters.    We thank the reviewer for their time and effort in reviewing our manuscript. 

A1: We agreed with the comment and have prepared a table (Supplemental Materials, Table S2) containing the exact search strings that we used for each database. Our search did not include any date, language, or publication type restrictions.

C2: There is a discrepancy between the text, which states that 39 articles were reduced to 34 reports, and the table, which still lists 39 literature cases. The authors should clarify whether this reduction reflects the merging of multiple publications referring to the same patient and update the table accordingly to ensure consistency and accurate representation of the dataset.

A2: We appreciate the reviewer's careful reading and constructive feedback. To address this discrepancy, we have clarified the distinction between 'studies' and 'cases' throughout the manuscript and updated the text for consistency. For example, we have revised the abstract to state: “… Results: We analyzed 34 studies with 39 individual cases. Additionally, this review was supplemented by one new illustrative case from our institution.” We also checked and corrected the text, all figures and tables to reflect these changes.

C3: The manuscript would benefit from either a Kaplan–Meier pooled survival analysis, if individual patient data are accessible, or the inclusion of a forest plot comparing survival outcomes across different treatment modalities such as surgery, radiosurgery, and WBRT with or without chemotherapy.

A3: Thank you for the suggestion. We have now included Kaplan-Meier survival analyses and forest plots to compare survival outcomes across the different treatment modalities. This new analysis confirms that while most individual treatment strategies did not show a statistically significant difference in survival, the inclusion of chemotherapy in any regimen was associated with a significant survival benefit (median survival 18.0 vs. 5.0 months, p = 0.04). A Cox proportional-hazards model further revealed that chemotherapy was associated with a roughly 70% reduction in the hazard of death. These figures and corresponding text have been incorporated into the manuscript to strengthen our findings.

C4: The inclusion of immunohistochemical panel positivity percentages adds value, yet the absence of key proliferative or molecular markers such as Ki-67, PIK3CA, or TP53 limits the depth of tumor characterization. If Ki-67 data were reported in any of the cases, the authors should include its range.

A4: Thank you for this valuable feedback. You are correct that a comprehensive analysis of proliferative markers is limited by the data available in the case reports. We have now revised the manuscript to include the specific Ki-67 data from the two publications that reported it, providing a clearer, albeit still limited, characterization of the tumor biology. We incorporated the following sentence in section 3.5. Histopathology (page 7): … While a comprehensive marker panel was often unavailable, the Ki-67 proliferation index was specifically reported in two instances, with values of approximately 20% and greater than 25%.

C5: Tables 1 and 2 are overly large, spanning multiple pages with over 40 columns in a font size that compromises readability, and the dense use of abbreviations further impedes clarity. The authors should restructure the data presentation by providing (i) a concise summary table outlining aggregate characteristics such as demographics and counts of key clinical or pathological variables, and (ii) a detailed per-patient table as a supplementary file. 

A5: We agree with the reviewer's feedback and have restructured the data presentation. A new summary tables have been included in the manuscript, and the original, detailed tables have been moved to supplementary materials.

C6: All figures should be exported at a minimum of 300 dpi, use a color-blind-safe palette, and employ vector graphics to ensure publication-quality resolution and accessibility.

A6: We have updated all figures according to the specifications. Plots now use a color-blind-safe palette (Viridis) and are provided as vector graphics (PDF), while all other images are at a minimum of 300 dpi. We would be happy to work with the production team if any further adjustments are needed.

C7: The manuscript contains numerous typographical errors (e.g., “symptems,” “inlcudes”), inconsistent verb tenses, and excessively long sentences, many exceeding 40 words, which collectively hinder readability. A professional copy-edit is necessary to correct these issues and improve overall fluency. The authors should aim to limit sentence length to 25 words or fewer to enhance clarity.

A7: Thank you for your valuable feedback. We have carefully proofread the manuscript and have done our best to correct all typographical errors and ensure verb tense consistency. We have also revised the text to shorten long sentences.

C8: While the flow diagram in Figure 1 adequately outlines the study selection process, the accompanying PRISMA 2020 checklist is missing. To comply with current reporting standards and ensure methodological transparency, the authors should complete and upload the PRISMA 2020 checklist as a supplementary file.

A8: We have now included the completed PRISMA 2020 checklist as a supplementary file as recommended.

Reviewer 2 Report

Comments and Suggestions for Authors

This review article comprehensively surveys previously reported cases of brain metastases from uterine leiomyosarcoma (ULMS) and presents the data as a case series. The authors are to be commended for compiling and organizing information on such a rare manifestation of an already rare disease, which has high academic value. Although my comments are largely minor, I offer the following suggestions for revision and clarification.

#1. Regarding survival data, I would note that survival should ideally be calculated using the Kaplan–Meier method. In the current study, cases that were "alive at the time of publication" appear to have been excluded from the analysis, which may have contributed to the extremely poor prognostic outcome reported. I recommend making every possible effort to include “alive at publication” cases in the survival analysis. If such inclusion is not feasible, the potential impact of this exclusion should be explicitly and repeatedly emphasized throughout the manuscript—in the Abstract, Results, Discussion, and Conclusions sections.

#2. In Table 1, the annotation for EOR appears to be misplaced and should more appropriately be attached to Table 2. I suggest reviewing all annotations to ensure their accuracy and correct placement.

#3. One valuable clinical lesson from this study is that clinicians should maintain a high index of suspicion for brain metastases in ULMS patients presenting with new neurological symptoms, especially if pulmonary metastases are present. In your current cohort, how many cases demonstrated a delay in brain imaging despite presenting with neurological symptoms? If such diagnostic delays were frequent, this would further strengthen the clinical significance of your work.

#4. Given the extremely poor prognosis observed in ULMS patients with brain metastases, the potential role of earlier detection through imaging-based screening warrants consideration. If you have any recommendations regarding the feasibility or value of such screening, I encourage you to add them.

#5. While the present study found no significant difference in prognosis according to treatment modality, it would be helpful to interpret these findings more precisely by presenting survival times stratified by treatment. I suggest adding a table and survival curves that show outcomes by treatment modality, and ideally also include other relevant factors such as patient age, presence of pulmonary metastases, and interval from initial diagnosis to the development of brain metastases.

#6. Finally, the References section lacks consistency and is insufficiently polished. Please review and revise for uniformity in formatting.

Author Response

C1: Regarding survival data, I would note that survival should ideally be calculated using the Kaplan–Meier method. In the current study, cases that were "alive at the time of publication" appear to have been excluded from the analysis, which may have contributed to the extremely poor prognostic outcome reported. I recommend making every possible effort to include “alive at publication” cases in the survival analysis. If such inclusion is not feasible, the potential impact of this exclusion should be explicitly and repeatedly emphasized throughout the manuscript—in the Abstract, Results, Discussion, and Conclusions sections.

A1: We thank the reviewer for their thoughtful feedback.

To clarify, we did not exclude all cases where patients were "alive at the time of publication." Instead, to ensure the most accurate estimate possible, we utilized all available data. For several cases where a diagnosis date was reported, we calculated the duration from diagnosis of brain metastasis to the last recorded follow-up. We agree that this approach contributes to an underestimation of overall survival, and we have added a note to the manuscript to explicitly address this limitation (3.2 Patient Characteristics, page 5).

A similar comment about survival assessment was raised by Reviewer 1. We have also included Kaplan-Meier survival analyses and forest plots to compare survival outcomes across the different treatment modalities. This new analysis confirms that while most individual treatment strategies did not show a statistically significant difference in survival, the inclusion of chemotherapy in any regimen was associated with a significant survival benefit (median survival 18.0 vs. 5.0 months, p = 0.04). A Cox proportional-hazards model further revealed that chemotherapy was associated with a roughly 70% reduction in the hazard of death. These figures and corresponding text have been incorporated into the manuscript to strengthen our findings.

C2: In Table 1, the annotation for EOR appears to be misplaced and should more appropriately be attached to Table 2. I suggest reviewing all annotations to ensure their accuracy and correct placement.

A2: Thank you for your feedback. We've corrected Table 1 as per the reviewer's comments and have ensured the EOR annotation, along with all others throughout the document, is correctly placed.

C3: One valuable clinical lesson from this study is that clinicians should maintain a high index of suspicion for brain metastases in ULMS patients presenting with new neurological symptoms, especially if pulmonary metastases are present. In your current cohort, how many cases demonstrated a delay in brain imaging despite presenting with neurological symptoms? If such diagnostic delays were frequent, this would further strengthen the clinical significance of your work.

A3: Thank you again for your valuable feedback. We agree that a high index of suspicion for brain metastases in patients with new neurological symptoms is a crucial clinical lesson. We have reviewed our cohort data and found that none of the cases reported a significant delay in brain imaging following the onset of neurological symptoms. This is likely due to the established clinical guidelines and standard of care practices in the institutions where these patients were treated. 

C4: Given the extremely poor prognosis observed in ULMS patients with brain metastases, the potential role of earlier detection through imaging-based screening warrants consideration. If you have any recommendations regarding the feasibility or value of such screening, I encourage you to add them.

A4: We agree that the extremely poor prognosis of brain metastases in uterine leiomyosarcoma (ULMS) patients highlights the need for earlier detection. While a comprehensive imaging-based screening strategy for all ULMS patients is not currently supported by evidence and may not be feasible due to cost and radiation exposure, we believe a more targeted approach is warranted.

We recommend that clinicians maintain a high index of suspicion for brain metastases in ULMS patients, particularly those with confirmed pulmonary metastases, as this is a common pathway for disease spread. We propose a strategy of close, regular neurological examinations for this high-risk subset of patients. Furthermore, we advise that brain imaging, such as an MRI, be performed promptly at the first sign of even minimal neurological changes. This proactive approach, while not a universal screening program, could significantly reduce diagnostic delays and may allow for earlier intervention, potentially improving patient outcomes.

We have incorporated these recommendations into the Discussion section (page 13). Thank you for your suggestion.

C5: While the present study found no significant difference in prognosis according to treatment modality, it would be helpful to interpret these findings more precisely by presenting survival times stratified by treatment. I suggest adding a table and survival curves that show outcomes by treatment modality, and ideally also include other relevant factors such as patient age, presence of pulmonary metastases, and interval from initial diagnosis to the development of brain metastases.    

A5: A similar comment about survival assessment was raised by Reviewer 1. We have now included Kaplan-Meier survival analyses and forest plots to compare survival outcomes across the different treatment modalities. This new analysis confirms that while most individual treatment strategies did not show a statistically significant difference in survival, the inclusion of chemotherapy in any regimen was associated with a significant survival benefit (median survival 18.0 vs. 5.0 months, p = 0.04). A Cox proportional-hazards model further revealed that chemotherapy was associated with a roughly 70% reduction in the hazard of death. These figures and corresponding text have been incorporated into the manuscript to strengthen our findings.

C6: Finally, the References section lacks consistency and is insufficiently polished. Please review and revise for uniformity in formatting.    

A6: Thank you for your feedback. We have double-checked all references and revised the References section to ensure uniformity and correct formatting, in accordance with the specified guidelines.

Round 2

Reviewer 1 Report

Comments and Suggestions for Authors

The authors have addressed all of my comments.